# Impact of Land Use Change on Carbon Storage Based on FLUS-InVEST Model: A Case Study of Chengdu–Chongqing Urban Agglomeration, China

Zhouling Shao [1], Chunyan Chen [1], Yuanli Liu [1], Jie Cao [1], Guitang Liao [2] and Zhengyu Lin [1,*]

1 Institute of Agricultural Information and Rural Economy, Sichuan Academy of Agricultural Science, Chengdu 610066, China; shaozl612@163.com (Z.S.); chenchunyan2000@126.com (C.C.); lyl757@163.com (Y.L.); caoxjiec@163.com (J.C.)

2 College of Resources and Environment, Chengdu University of Information Technology, Chengdu 610225, China; lgt@cuit.edu.cn

* Correspondence: linzhengyu_saas@163.com

**Abstract:** Land use change is one of the main factors driving changes in terrestrial carbon storage, which comprises the storage of vegetation carbon and soil carbon. Selecting the Chengdu–Chongqing urban agglomeration (CCUA) as the study area, land use and carbon storage from 2010 to 2030 were analyzed by combining the Future Land Use Simulation (FLUS) model and the Integrated Valuation of Ecosystem Services and Tradeoffs (InVEST) model. The main types of land use in CCUA are farmland and forest. The conversion of farmland to built-up land was the most important form of land use transfer between 2010 and 2020. Each type of land use shows the smallest change under the ecological protection scenario, and the degree of the comprehensive land use dynamic is only 0.19%. Under the natural development scenario, the areas of built-up land, wetland, and forest land will increase in 2030. Under the urban development scenario, the built-up land area will increase by 751.24 km$^2$, an increase in more than 10.08%, but farmland, forest, and grassland will decrease. The spatial pattern of carbon storage is "high in the east and west, low in the middle"; farmland accounts for the largest proportion of carbon storage at over 60% of the total. Carbon storage decreased by 29.45 × 10$^6$ Mg from 2010 to 2020. Grassland showed the most significant decrease in carbon storage, with the proportion decreasing from 7.49% in 2010 to 6.09% in 2020. In 2030, the total carbon storage will reach 1844.68 × 10$^6$ Mg under the ecological protection scenario, slightly higher than that in 2020, while it will show a downward trend under the natural development and urban development scenarios.

**Keywords:** land use; carbon storage; spatio-temporal change; FLUS model; InVEST model; scenario; Chengdu–Chongqing urban agglomeration (CCUA)

## 1. Introduction

Terrestrial ecosystems play a key role in the global carbon cycle. Land use change is one of the most important factors driving changes in carbon storage in terrestrial ecosystems, and its contribution to the increase in $CO_2$ content in the atmosphere is second only to fossil energy [1–4]. Changes in the structure, function, and process of the land-use system lead to changes in the carbon storage of terrestrial ecosystems. Although decreases in terrestrial carbon storage can increase atmospheric $CO_2$ content, its increase can effectively reduce atmospheric $CO_2$ content [5–7]. Therefore, regulation of the land use system is considered one of the most economically feasible and environmentally friendly strategies for controlling $CO_2$ emissions, and has consistently been the policy focus of global governance and the frontier hotspot of land science [8–10]. The carbon-fixation capacity of different types of land use is different, and the land use cover type will directly affect the carbon-balance mechanism of terrestrial ecosystems [11]. In recent years, domestic and

foreign scholars have extensively investigated the relationship between land use change and carbon storage of terrestrial ecosystems, with two main characteristics.

In terms of research methods, studies have evolved from the early sample land-inventory method to the current combination of remote-sensing monitoring and modeling [12]. The traditional sample land inventory method requires large amounts of labor and material resources, the time cost for updating is very high, and it is more suitable for small-scale monitoring. In contrast, remote sensing provides the possibility for large-scale monitoring, and its combining with modeling techniques provides a cost-effective means of carbon assessment [13]. In addition, the InVEST model requires fewer data, ensures rapid processing, and has high accuracy. It can also realize the visual expression of the spatial distribution and dynamic change in carbon storage, and is now widely used in the estimation of carbon storage associated with land use [14,15]. Scenario simulations of future land use change are helpful in quantifying the direct relationship between carbon storage and future land use, and are of great significance for understanding the carbon sequestration of terrestrial ecosystems. At present, commonly used simulation methods include the CA-Markov, CLUE-S, and FLUS models [14,16,17]. Among these, the FLUS model is a land spatial-temporal simulation model that is further improved on the basis of the CA (cellular automata) model. It first selects the driving factors through the random forest decision tree, and then uses the system dynamics method to simulate the land use in future years, facilitating the proper handling of the relationship between different land use types [17]. The FLUS-InVEST model solves the problem of the traditional CA model being unable to evaluate the complex relationship between dynamic changes in land use and carbon storage, and has higher simulation accuracy than the traditional CA model [18,19].

Regarding the assessment scale, the study of carbon storage has been expanded from single ecosystem types such as forest, grassland, and farmland to river basins, provinces, typical ecological areas, and urban agglomerations [16,20–22]. Many studies have investigated the carbon emissions related to land use in large developed cities in East China, such as Beijing and Guangdong, and economically developed regions such as the Beijing-Tianjin-Hebei urban agglomeration, the Pearl River Delta urban agglomeration, and the Yangtze River Delta urban agglomeration [13,23–27]. The results show that the optimization of the land use structure is an effective means of reducing carbon emissions [28,29]. However, research on developing and relatively underdeveloped cities and regions in Western China is lacking. These areas in western China are significantly different from those in eastern China in terms of landform, climate, and socio-economic background [30].

Located in the upstream area of the Yangtze River, the Chengdu–Chongqing urban agglomeration (CCUA) is the ecological barrier of the Yangtze River Economic Belt and is representative of the mountainous urban agglomeration in the western region [31]. With its unique geographical location, CCUA is a major national strategic deployment. In the past decades, it has experienced rapid population growth and significant changes in land use, posing a serious threat to ecosystem carbon storage services [32,33]. Therefore, the study of CCUA would improve our understanding of how land use change affects carbon storage in the western region. At the same time, the quantitative assessment and prediction of the impact of land use change on the carbon storage of the ecosystem in CCUA, and exploring means of optimizing the land use structure, will help to improve the carbon storage of the regional ecosystem. This is also of great significance for improving the regional ecosystem service function and mitigating climate change [34]. The general objective of this study was to evaluate the impact of spatio-temporal changes in land use on carbon storage in CCUA. The specific objectives were (I) to quantify and map the temporal and spatial patterns of land use and carbon storage in different periods using land use images from 2010, 2015, and 2020; (II) to simulate the future land use changes and their impact on carbon storage in 2030 under multiple scenarios, taking into account the natural environment and socio-economic driving factors.

## 2. Data Source and Research Methods

### 2.1. Study Area

CCUA is located in Southwest China (101°57′56″ E–108°56′40″ E, 27°40′31″ N–32°19′21″ N) (Figure 1). It borders Hubei in the east, and Yunnan and Guizhou in the south, comprising 15 cities in Sichuan Province and 29 districts and counties in Chongqing, with a total area of approximately 185,000 km². CCUA has a subtropical monsoon climate, with annual average temperature of 16–18 °C and annual precipitation of 900–1200 mm. It is located in the Sichuan Basin and is upstream of the Yangtze River, with an altitude of 71–5834 m. The terrain is relatively low in the middle and surrounded by mountains in the southwest and east, mainly including plains, hills, and mountains. The highest elevation in CCUA is represented by the Hengduan Mountains in the western part, and the eastern part comprises the parallel mountains and valleys in the east of Sichuan. With dense river network and developed irrigation water system, the area enjoys the reputation of being a "Land of Abundance". The Yangtze River runs through the south of the region from west to east, and important rivers such as the Wujiang River, Minjiang River, Jialing River, and Dadu River also flow through the study area.

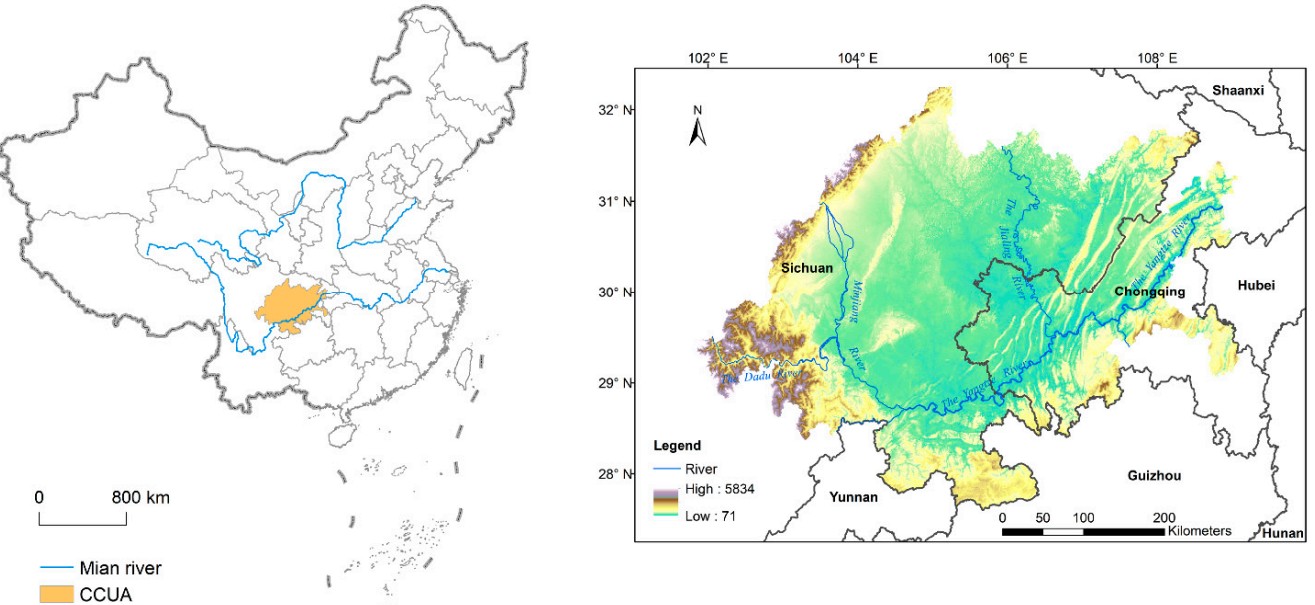

**Figure 1.** Geographic location of CCUA.

The area includes the important soil conservation in the Three Gorges Reservoir Area, the area of water conservation and biodiversity conservation in Daluoshan District, the area of biodiversity conservation and water conservation in Wuling Mountain Area, the important area of biodiversity conservation and water conservation in Minshan-Qionglai-Liangshan Mountains, and other important areas of national ecological functions. It undertakes the functions of ecological protection such as water and soil conservation, water conservation, and biodiversity conservation. In addition, as the ecological protection belt in the upstream of the Yangtze River, the largest urban agglomeration in the western region, CCUA also has the regional advantages of connecting the east to the west and connecting the north and the south in China [31].

### 2.2. Data Source and Preprocessing

Data mainly included land use, restricted conversion area data, carbon density data, and land use change drivers (Table 1). Land use data were mainly obtained through manual visual interpretation of Landsat remote sensing images with a spatial resolution of 1 km. This study used four phases of data in 2010, 2015, 2018, and 2020. ArcGIS software was used to reclassify the original land use types into six categories: farmland, forest land, grassland,

built-up land, wetland, and other land (Table 2). The data of restricted conversion area were derived from vector data of nature reserves in China. After converting them to raster data, the nature reserves of the study area were isolated, and the values of non-protected and protected areas were assigned 1 and 0, respectively, as input parameters of the restricted transformation area. The carbon density data were obtained from the National Ecological Science Data Center [35]. By selecting the data of areas in or near the study area as the reference value of carbon density, the mean value was used to replace multiple data.

**Table 1.** Datasets used in this study.

| Data Attribute | Resolution | Data Source |
| --- | --- | --- |
| LUCC (2010, 2015, 2018, 2020) | 1 km | RESDC |
| Natural reserve | — | RESDC |
| Carbon density | — | National Ecological Science Data Center (http://www.cnern.org.cn, accessed on 10 December 2022) |
| Elevation | 30 m | Geospatial Data Cloud (http://www.gscloud.cn, accessed on 10 December 2022) |
| Soil type | 1 km | RESDC |
| Slope | 1 km | — |
| Temperature | 1 km | National Meteorological Science Data Center (http://data.cma.cn, accessed on 10 December 2022) |
| Precipitation | 1 km | National Meteorological Science Data Center |
| Normalized Difference Vegetation Index (NDVI) | 1 km | RESDC |
| Night-time satellite data | 1 km | RESDC |
| Gross domestic product (GDP) | 1 km | RESDC |
| Population | 1 km | RESDC |
| Distance to road | 1 km | OSM (https://www.openstreetmap.org/, accessed on 12 December 2022) |
| Distance to rail transport | 1 km | OSM |
| Distance to river | 1 km | OSM |

**Table 2.** Land use classification.

| | Sub-Categories |
| --- | --- |
| 1 Farmland | Paddy field |
| | Dry farmland |
| 2 Forest land | Wood land |
| | Shrub land |
| | Sparsely forested land |
| | Other forested land |
| 3 Grassland | High coverage grassland |
| | Middle coverage grassland |
| | Low coverage grassland |
| 4 Wetland | River and canal |
| | Lake |
| | Reservoir and waterhole |
| | Tidal marsh |
| | Shoal and reed land |
| 5 Built-up land | Cities and towns |
| | Rural settlements |
| | Industry and traffic land |
| 6 Other land | Sandy land |
| | Gobi |
| | Saline-alkali land |
| | Swampland |
| | Bare land |
| | Rock and gravel |
| | Other unused land |

In total, 12 driving factors of land use change were considered, including socio-economic data and environmental data. Among them, elevation data were obtained from the GDEM-V2 data provided by the NASA Earth Observation Satellite Terra, with a resolution of 30 m. Its resolution was unified with the land use data using the resampling tool. In this study, the latest road and water system data were used, and the Euclidean distance was calculated through ArcGIS software, and raster data with a resolution of 1 km were obtained. NDVI, GDP, population kilometer grid, night-time satellite data, and other raster data were obtained from the Resource and Environment Science and Data Center (RESDC https://www.resdc.cn/, accessed on 10 December 2022).

*2.3. Research Methods*

2.3.1. Method Framework

This study combined the FLUS model and the InVEST model to analyze and simulate carbon storage and its change in the terrestrial ecosystem of CCUA (Figure 2). First, based on land use data and carbon density data, the change in land use and carbon storage in CCUA from 2010 to 2020 was analyzed. Then, combined with historical land use change patterns, the Markov model was used to predict the number of future land use changes. Using the driving factors of land use change and the data of the restricted zone, the FLUS model was used to simulate land use change under three scenarios of natural development, ecological protection, and urban development. Finally, the InVEST model was used to predict the spatial distribution characteristics and changes in carbon storage under three scenarios in 2030, and to explore the impact of land use change on carbon storage in CCUA.

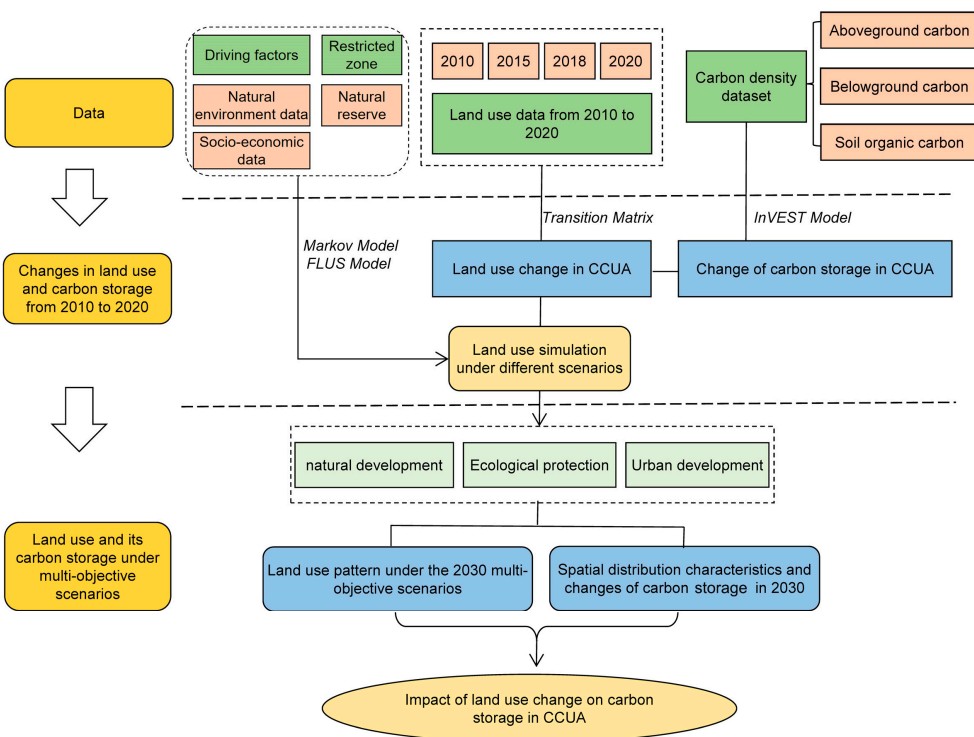

**Figure 2.** Research framework.

2.3.2. FLUS Model and Multi-Scenario Setting

The FLUS model was further improved on the basis of the traditional CA model. The model includes a fitness probability calculation, neighborhood factor calculation, adaptive inertia coefficient calculation, conversion cost setting, and comprehensive probability calculation [36–38]. The suitability probability calculation uses the BP-ANN artificial neural network algorithm to fit the land use types in the base period with multiple spatial driving factors, so as to obtain the suitability probability of each land use type [39]. The spatial

computational simulation of the CA model is based on the adaptive inertial competition mechanism of roulette wheel selection [17]. It uses the land use data of the base year and the constructed probability atlas of suitability to predict the spatial pattern of future land use. It can solve the uncertainty and complexity of mutual conversion of various land use types under the common influence of natural action and human activities, and calculate comprehensive rules.

Based on the land use data of CCUA from 2010 to 2020, and the driving factor data of land use change, the suitability probability of various land types in the study area was obtained through the artificial neural network (ANN) algorithm module in the FLUS model. The CA based on the adaptive inertia mechanism was used to simulate the land use pattern in 2030 under multi-objective scenarios. Multi-objective scenarios refer to the implementation of different measures according to different development needs. The natural development scenario is based on historical land use change rate and natural and human driving factors. It does not consider the policy macro-control, nor does it change the change rate and conversion rules between different land use types. According to the existing land use transfer matrix, future land use change was simulated under the natural development scenario. With the proposed restoration of the overall protection system of mountains, rivers, forests, fields, lakes, and grasslands, ecological civilization has received widespread attention. Therefore, the ecological protection scenario restricts the transfer of ecological land and the development of urban built-up land. Under the urban development scenario, the impact of economic and social benefits on the mode of future land use was considered [40].

The neighborhood weight is the expansion intensity of the land use type, and the larger the value, the stronger its expansion. This study obtained neighborhood weights based on the proportion of the expansion area of each land use type from 2018 to 2020 (Table 3). This was also achieved by analyzing historical land use changes and setting neighborhood weights for the FLUS model. In this manner, bias caused by subjective judgments could be reasonably avoided, and the objectivity and scientificity of the simulation could be improved. The cost conversion matrix represents the difficulty of converting a land type to other land use types, and "0" represents land that cannot be converted. Different cost conversion matrices were set based on three scenarios (Table 4).

**Table 3.** Neighborhood weight of each land use type.

| Land Use Type | Farmland | Forest Land | Grassland | Wetland | Built-Up Land | Other Land |
| --- | --- | --- | --- | --- | --- | --- |
| Neighborhood weight | 0.0073 | 0.0245 | 0.2006 | 0.04455 | 0.5973 | 0.1248 |

**Table 4.** Land use transfer cost matrix under three scenarios.

| Scenario | LUCC Type | Farmland | Forest Land | Grassland | Wetland | Built-Up Land | Other Land |
| --- | --- | --- | --- | --- | --- | --- | --- |
| Natural development | Farmland | 1 | 1 | 1 | 1 | 1 | 1 |
| | Forest land | 1 | 1 | 0 | 0 | 0 | 0 |
| | Grassland | 1 | 1 | 1 | 1 | 1 | 1 |
| | Wetland | 0 | 0 | 1 | 1 | 1 | 1 |
| | Built-up land | 1 | 0 | 1 | 1 | 1 | 1 |
| | Other land | 1 | 1 | 1 | 0 | 1 | 1 |
| Ecological protection | Farmland | 1 | 1 | 1 | 1 | 0 | 1 |
| | Forest land | 0 | 1 | 0 | 0 | 0 | 0 |
| | Grassland | 0 | 1 | 1 | 1 | 0 | 0 |
| | Wetland | 0 | 0 | 0 | 1 | 0 | 0 |
| | Built-up land | 0 | 0 | 0 | 0 | 1 | 0 |
| | Other land | 1 | 1 | 1 | 1 | 1 | 1 |
| Urban development | Farmland | 1 | 0 | 0 | 0 | 0 | 1 |
| | Forest land | 1 | 1 | 1 | 0 | 0 | 0 |
| | Grassland | 1 | 1 | 1 | 1 | 1 | 1 |
| | Wetland | 1 | 0 | 1 | 1 | 0 | 1 |
| | Built-up land | 0 | 0 | 0 | 0 | 1 | 0 |
| | Other land | 1 | 1 | 1 | 0 | 1 | 1 |

The total conversion probability of a specific land type was calculated by combining the predicted quantity, suitability probability, neighborhood weight, adaptive inertia coefficient, conversion cost, and restricted area. The expression is:

$$Tp_{g,k}^t = sg(g,k,t) \times \Omega_{g,k}^t \times I_k^t \times \left(1 - sp_{p \to k}\right)$$

where $Tp_{g,k}^t$ is the total probability of land type $g$ to land type $k$ at time $t$; $sp_{p \to k}$ is the cost of converting $p$ types of land use into type $k$ in which $\left(1 - sp_{p \to k}\right)$ represents the difficulty of transformation; and $\Omega_{g,k}^t$ is a neighborhood function.

The Kappa coefficient was used to evaluate the predictive accuracy of the model. The Kappa coefficient evaluation criteria were 0.80–1.00 for excellent prediction, 0.60–0.80 for good prediction, 0.40–0.60 for fair prediction, and 0–0.40 for poor prediction:

$$\text{Kappa} = \frac{P_0 - P_e}{1 - P_e}$$

where $P_0$ is the overall simulation accuracy and $P_e$ is the theoretical simulation accuracy.

Considering the data for 2015 is far from today, when analyzing the land use change patterns of CCUA from 2010 to 2020, it is more effective to select a more recent year in areas with rapid urban development and severe land change to truly reflect the situation of land use change. Therefore, this study selected the land use data in 2018 to simulate the land use situation in 2020, and the simulated data were compared with the actual value in 2020. In this manner, the prediction accuracy of the model was determined. The results showed that the overall accuracy of land use data in 2020 simulated by the FLUS model was 91.18%, and the Kappa coefficient was 83.71%. These values indicate that the FLUS model has good simulation ability and high accuracy in land use prediction in CCUA and can be used to simulate land use change in this region.

### 2.3.3. InVEST Model and Carbon Density

InVEST is a suite of models used to map and value the goods and services from nature that sustain and fulfill human life. Carbon storage refers to the amount of carbon stored in a carbon pool, such as forests, oceans, and land. Carbon density is the carbon storage per unit area [14,18]. Based on the land use data and the carbon storage of the four carbon pools, the carbon module in the InVEST model was used to estimate the carbon storage in the current ecosystem or in a period of time [41]. The carbon density data of CCUA are presented in Table 5. The model divides the ecosystem carbon pool into four basic carbon pools: aboveground biological carbon, underground biological carbon, soil organic carbon, and dead organic carbon [15]. As the carbon density of dead organic matter is small and relevant data are difficult to obtain, the carbon pool was not considered in this study. The carbon storage of terrestrial ecosystems is the sum of the total carbon density of each land use type multiplied by the corresponding area. The calculation formula is as follows:

$$C_i = C_{above} + C_{below} + C_{soil} + C_{dead}$$

$$C_{total} = \sum_{i=1}^{n} C_i \times S_i$$

where $C_i$ is the overall carbon density; $C_{above}$, $C_{below}$, $C_{soil}$, and $C_{dead}$ are the density of aboveground biological carbon, underground biological carbon, soil organic carbon, and dead organic carbon, respectively; $C_{total}$ is the total carbon storage; and $S_i$ is the area of land use type $i$.

**Table 5.** Reference values of land use carbon density in the study area (Mg/hm$^2$).

| Lucode | Land Use Type | $C_{above}$ | $C_{below}$ | $C_{soil}$ | $C_{total}$ |
|---|---|---|---|---|---|
| 1 | Farmland | 21.83 | 13.64 | 68.24 | 103.71 |
| 2 | Forest land | 44.38 | 9.35 | 51.9 | 105.63 |
| 3 | Grassland | 18.37 | 21.43 | 76.56 | 116.36 |
| 4 | Wetland | 0 | 0 | 0 | 0 |
| 5 | Built-up land | 0.71 | 1.34 | 33.99 | 36.04 |
| 6 | Other land | 9.13 | 1.82 | 34.08 | 45.03 |

## 3. Results and Analysis

### 3.1. Analysis of Land Use Change

As shown in Figure 3A–C, the main land use types of CCUA in 2010, 2015, and 2020 were farmland and forest land. In 2020, farmland accounted for 61.76% of the total land use in the region, while forest land accounted for 27.12%. Grassland and built-up land accounted for 5.24% and 4.03%, respectively. The area of wetland and other land use types was less than 2.00% of the land use area of the whole region. In terms of spatial distribution, farmland was mainly distributed in the flat plain and the eastern hilly area. Built-up land spread outward from Chengdu–Chongqing as the center. In addition, built-up land was distributed in the form of dot clusters in other areas, forming towns of different sizes. Forest land was mainly distributed in the southwestern part of the study area and the eastern mountainous area with relatively high elevation. From the perspective of area change, all land use types exhibited distinct changes during 2010–2020. Specifically, the area of built-up land, wetland, forest land, and other land types continuously expanded, whereas the area of grassland and farmland continuously shrunk. With reference to 2010, built-up land exhibited the most drastic growth, with an increase in 82.22%. Grassland area presented the most prominent decrease (20.21%). The changes in land use during 2010–2020 were mainly concentrated in the last five years, and the changes in farmland and forest land were the most significant.

Based on the analysis of land use change in CCUA from 2010 to 2020, a chord diagram was generated (Figure 3D) and a land use transfer matrix was constructed (Table 6) to show the detailed situation of land use change more intuitively. Through the analysis of the characteristics of land use transfer change in the study area from 2010 to 2020, farmland was found to mainly flow into forest land and grassland, and the outflow area accounted for 16.85% of the farmland area. Among built-up land, farmland was the main type of land inflow. The area of farmland converted into built-up land was 4496.97 km$^2$, accounting for 84.76% of the net inflow area of built-up land. The results indicate that, in the last 10 years, driven by economic development and urban expansion, a large amount of urban development has occupied the surrounding farmland and continued to expand in the periphery. In the study area, the main inflow of forest land was farmland, which was 15,128.02 km$^2$, accounting for up to 83.60% of the total transfer amount of forest land.

**Table 6.** Land use transfer matrix of CCUA from 2010 to 2020 (km$^2$).

| | Farmland | Forest Land | Grassland | Wetland | Built-Up Land | Other Land | 2010 |
|---|---|---|---|---|---|---|---|
| Farmland | 92,179.39 | 15,128.02 | 2976.81 | 1710.68 | 4496.97 | 20.16 | 116,512.03 |
| Forest land | 15,358.86 | 30,935.47 | 2208.67 | 275.20 | 446.57 | 81.65 | 49,306.42 |
| Grassland | 3909.27 | 3640.12 | 4400.20 | 72.58 | 111.90 | 18.15 | 12,152.22 |
| Wetland | 1365.93 | 237.90 | 52.42 | 977.82 | 246.98 | 12.10 | 2893.15 |
| Built-up land | 1501.01 | 210.69 | 42.34 | 183.47 | 2154.23 | 2.02 | 4093.76 |
| Other land | 33.27 | 52.42 | 16.13 | 14.11 | 3.02 | 60.48 | 179.43 |
| 2020 | 114,347.73 | 50,204.62 | 9696.57 | 3233.86 | 7459.67 | 194.56 | 185,137.01 |

NOTE: The data in each row are added to obtain the total area of land use in the same category in 2010, and the data in each column are added to obtain the total area of land use in the same category in 2020. The data in each cell represent the area of the row of land use type transferred to the column of land use type from 2010 to 2020. The total area of the study area is 185,137.01 km$^2$.

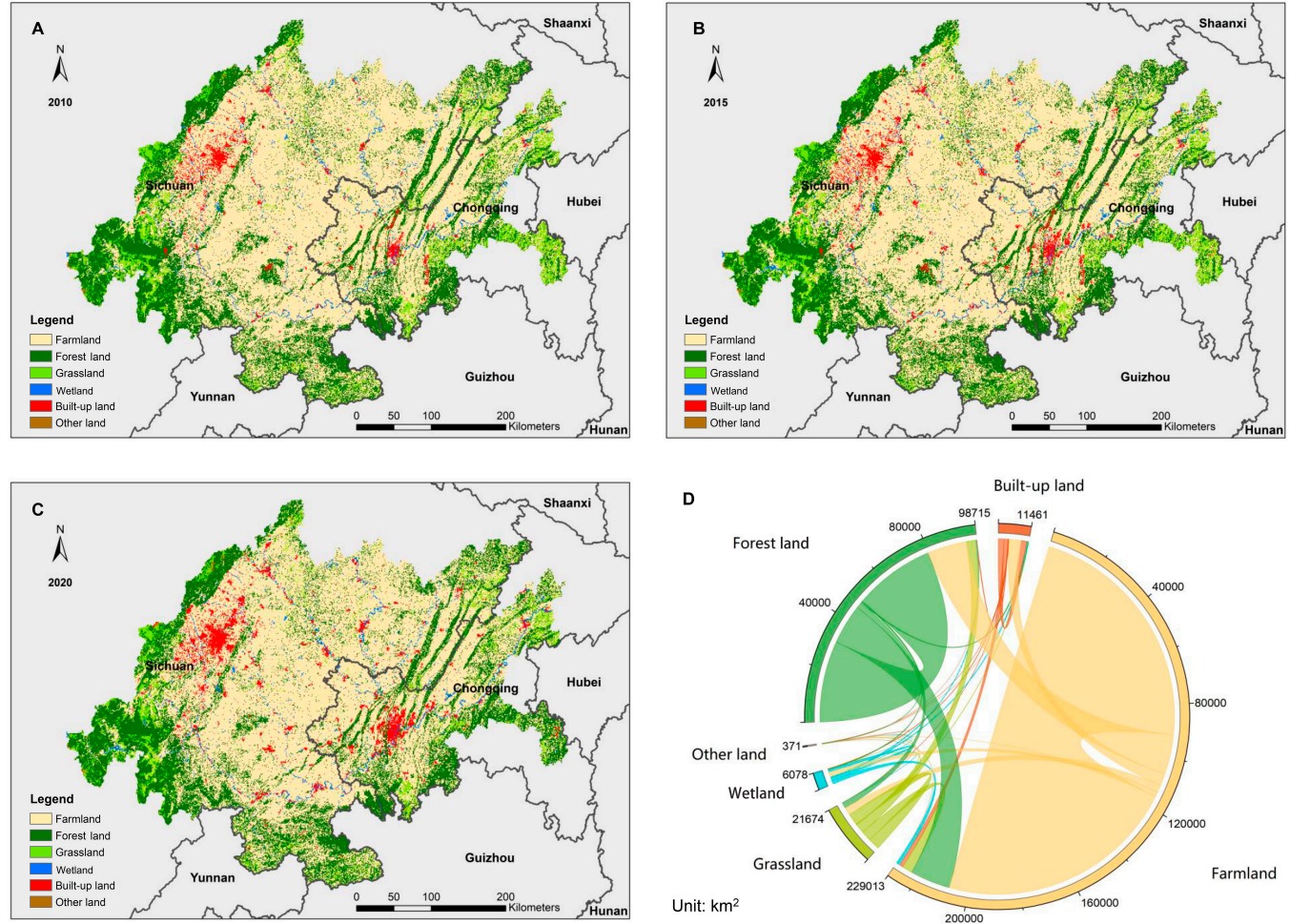

**Figure 3.** Land use spatial pattern and change in CCUA from 2010 to 2020. (**A–C**) represent the land use patterns of CCUA in 2010, 2015, and 2020, respectively. (**D**) shows the changes in land use from 2010 to 2020.

### 3.2. Land Use Prediction under Different Scenarios in the Future

By comparing the multi-scenario land use simulation results in Table 7 and Figure 4 with the land use change from 2010 to 2020, the overall pattern of land use in CCUA was found to be highly consistent, but local changes were prominent. Under the natural development scenario (Figure 4A), natural environmental factors did not exhibit any sudden change, and the current speed of social development was roughly maintained. Compared with the change trend of various types of land use in the previous decade, the dynamic attitude of comprehensive land use will decrease from 5.01 to 0.84% in 2030, indicating that the development of CCUA would be stable in the future, and the fluctuation of different types would be relatively small. During the period, built-up land, wetland, and forest land increased, among which built-up land increased by 651.54 km² and wetland increased by 114.80 km². The area of grassland and farmland decreased, among which grassland decreased by 533.72 km² and the farmland decreased by 236.65 km². Under the ecological protection scenario (Figure 4B), the change range of land use types was the smallest, and the dynamic attitude of comprehensive land use was only 0.19%. Only the areas of forest land and grassland showed a slight increasing trend, with a total increase of 180.26 km², while the areas of farmland, wetland, built-up land, and other types of land showed a downward trend. Under the urban development scenario (Figure 4C), the area of built-up land will increase by 751.24 km², an increase of more than 10.08%, while the development of urban and industrial and mining land will reduce the area of farmland,

forest land, and grassland by 749.22 km$^2$. The small area of forest land and grassland would pose a certain threat to the sustainable development of the region.

**Table 7.** Area and proportion of land use types under different scenarios in 2030.

| Land Use Type | Attribute | 2010 | 2020 | Natural Development | Ecological Protection | Urban Development |
|---|---|---|---|---|---|---|
| Farmland | Area (km$^2$) | 116,488.39 | 114,282.92 | 114,046.27 | 114,192.29 | 114,046.27 |
| | Rate (%) | 62.92 | 61.73 | 61.60 | 61.68 | 61.60 |
| Forest land | Area (km$^2$) | 49,336.91 | 50,255.44 | 50,263.50 | 50,360.17 | 50,002.68 |
| | Rate (%) | 26.65 | 27.15 | 27.15 | 27.20 | 27.01 |
| Grassland | Area (km$^2$) | 12,149.23 | 9718.77 | 9185.05 | 9794.30 | 9458.96 |
| | Rate (%) | 6.56 | 5.25 | 4.96 | 5.29 | 5.11 |
| Wetland | Area (km$^2$) | 2892.58 | 3233.55 | 3348.35 | 3228.51 | 3233.55 |
| | Rate (%) | 1.56 | 1.75 | 1.81 | 1.74 | 1.75 |
| Built-up land | Area (km$^2$) | 4090.69 | 7451.96 | 8103.51 | 7370.40 | 8203.20 |
| | Rate (%) | 2.21 | 4.03 | 4.38 | 3.98 | 4.43 |
| Other land | Area (km$^2$) | 179.21 | 194.36 | 190.33 | 191.33 | 192.34 |
| | Rate (%) | 0.10 | 0.10 | 0.10 | 0.10 | 0.10 |

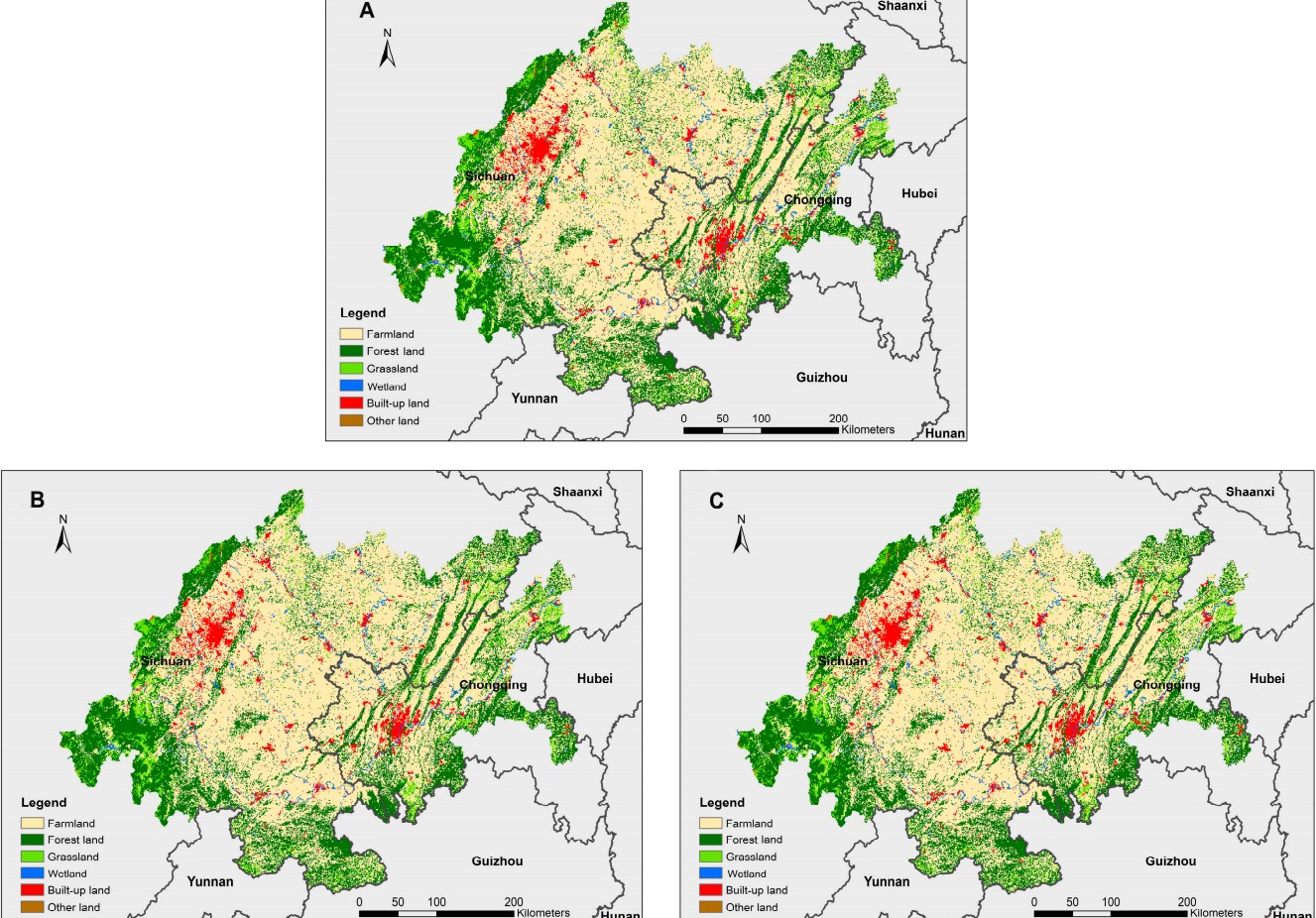

**Figure 4.** Land use simulation with multiple scenarios in 2030: natural development (**A**), ecological protection (**B**), and urban development (**C**).

### 3.3. Spatial Distribution Characteristics and Changes in Carbon Storage

In terms of spatial distribution (Figure 5), carbon storage was mainly distributed in the mountainous areas around the Sichuan Basin, such as the western and eastern parts of CCUA and parallel ridge valleys in eastern Sichuan. All carbon storage values in these regions were higher than 11,636 Mg per pixel. The lowest carbon storage was found in the river area, followed by built-up land in the central plain area of the study area. The carbon storage in this area was 3604 Mg per pixel. Meanwhile, there were two low-carbon reserve areas in the northwest and southeast of the Sichuan Basin, namely the main urban areas of Chengdu and Chongqing. The land types were mainly built-up land. From 2010 to 2020, the carbon storage of CCUA showed a downward trend as a whole, and the carbon storage of the region decreased by $29.45 \times 10^6$ Mg during the decade.

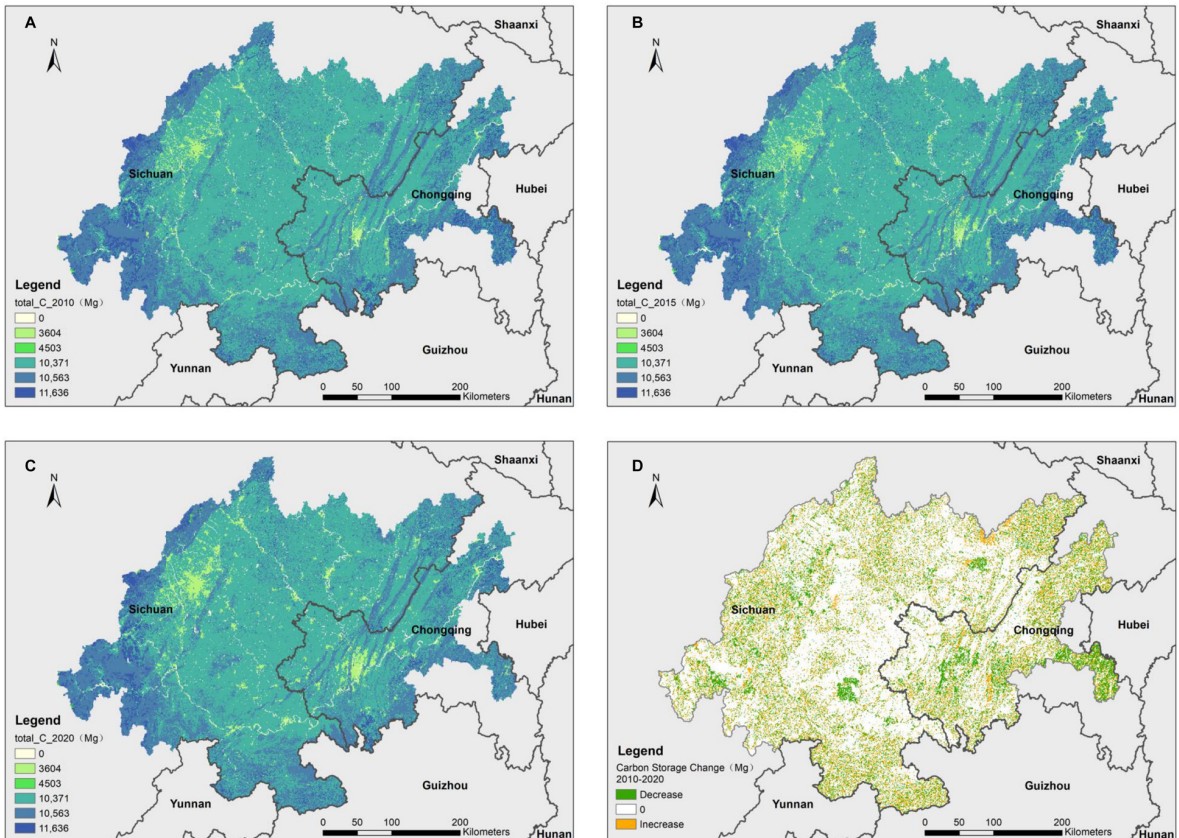

**Figure 5.** Spatial and temporal changes in carbon storage in CCUA from 2010 to 2020. (**A–C**) represent the carbon storage patterns of CCUA in 2010, 2015, and 2020, respectively. (**D**) shows the changes carbon storage from 2010 to 2020.

The prediction of land use carbon storage in CCUA under multiple scenarios in 2030 is shown in Figure 6. Based on the multi-scenario land use pattern in 2030, predicted using the FLUS model, the InVEST model was used to calculate carbon storage under the three scenarios. The results showed that, compared with 2020, the total carbon storage remained the same under the ecological protection scenario, while the total carbon storage showed a downward trend under the natural development and urban development scenarios. Under the natural development scenario (Figure 6A), the carbon storage of CCUA reached $1837.74 \times 10^6$ Mg, reflecting the largest decrease with reference to the total carbon storage in 2020. Under the ecological protection scenario (Figure 6B), the carbon storage of CCUA reached $1844.68 \times 10^6$ Mg, reflecting an increase by $0.73 \times 10^6$ Mg compared with the total carbon storage in 2020. This is primarily attributable to the ecological protection measures adopted under this scenario, which limits the reduction in ecological land. Under the urban development scenario (Figure 6C), the carbon storage of CCUA decreased by $5.41 \times 10^6$ Mg.

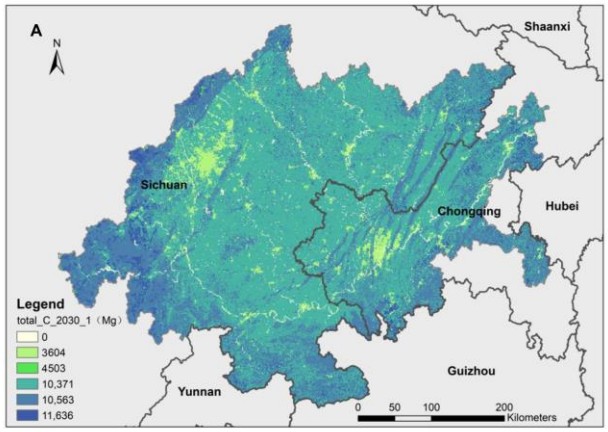

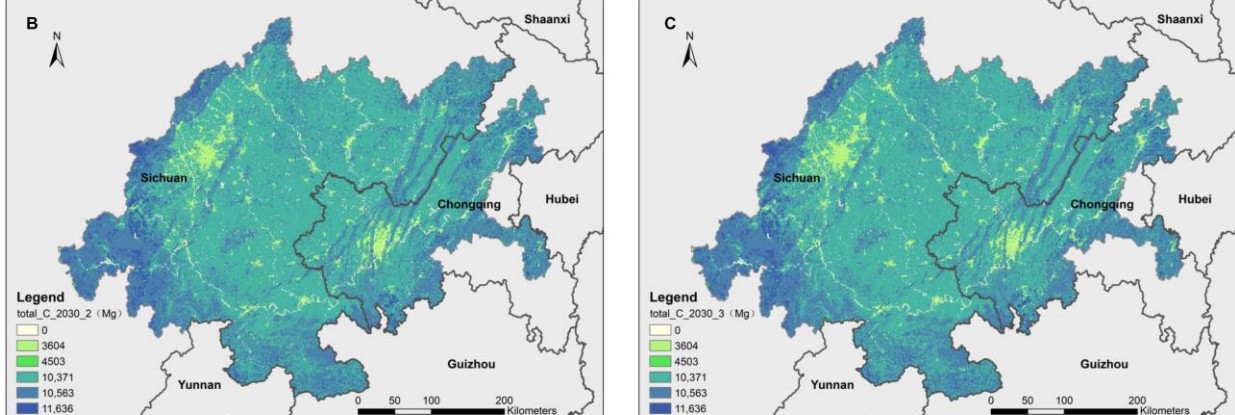

**Figure 6.** Carbon storage of CCUA under multiple scenarios in 2030: natural development (**A**), ecological protection (**B**), and urban development (**C**).

### 3.4. Effects of Land Use Types on Carbon Storage

Considering different land types, the influences on carbon storage were analyzed (Table 8). The contribution of farmland to the carbon storage of CCUA was the largest, followed by forest land and grassland. The relatively large carbon storage of farmland can be attributed to the flat terrain in the central part of the study area, which is rich in purple soil, paddy soil, and yellow soil. The superior natural environment and historical farming habits have promoted the wide distribution of farmland in the region, making it an important part of carbon storage. Owing to their low carbon density, the contribution of built-up land and other land types to carbon storage was low in the region.

**Table 8.** Carbon storage of land use from 2010 to 2030.

| Scenario Setting | 2010 | | 2015 | | 2020 | | Natural Development | | Ecological Protection | | Urban Development | |
|---|---|---|---|---|---|---|---|---|---|---|---|---|
| | Carbon Storage (10⁶ Mg) | Rate(%) | Carbon Storage (10⁶ Mg) | Rate(%) | Carbon Storage (10⁶ Mg) | Rate (%) | Carbon Storage (10⁶ Mg) | Rate (%) | Carbon Storage (10⁶ Mg) | Rate (%) | Carbon Storage (10⁶ Mg) | Rate (%) |
| Farmland | 1199.92 | 64.05 | 1186.53 | 63.63 | 1176.96 | 63.83 | 1174.53 | 63.91 | 1176.03 | 63.75 | 1174.53 | 63.88 |
| Forest land | 517.62 | 27.63 | 517.37 | 27.74 | 527.15 | 28.59 | 527.23 | 28.69 | 528.25 | 28.64 | 524.5 | 28.53 |
| Grassland | 140.41 | 7.49 | 140.38 | 7.53 | 112.3 | 6.09 | 106.13 | 5.78 | 113.17 | 6.13 | 109.3 | 5.94 |
| Wetland | 0 | 0.00 | 0 | 0.00 | 0 | 0.00 | 0 | 0.00 | 0 | 0.00 | 0 | 0.00 |
| Built-up land | 14.64 | 0.78 | 19.68 | 1.06 | 26.67 | 1.45 | 29 | 1.58 | 26.38 | 1.43 | 29.36 | 1.60 |
| Other land | 0.8 | 0.04 | 0.81 | 0.04 | 0.87 | 0.05 | 0.85 | 0.05 | 0.86 | 0.05 | 0.86 | 0.05 |
| Total | 1873.4 | 100.00 | 1864.76 | 100.00 | 1843.95 | 100.00 | 1837.74 | 100.00 | 1844.68 | 100.00 | 1838.54 | 100.00 |

From 2010 to 2020, the carbon storage of CCUA showed a downward trend. Among them, carbon storage was 1873.40 × 10⁶ Mg in 2010 and 1864.76 × 10⁶ Mg in 2015. The change in carbon storage from 2010 to 2015 is primarily attributable to the decrease in

farmland and the increase in built-up land, with farmland conversion accounting for a decrease of 13.39 $\times$ $10^6$ Mg in carbon storage. In 2020, carbon storage was further reduced by 20.81 $\times$ $10^6$ Mg compared with 2015. The significant reduction in grassland area resulted in a reduction of 28.08 $\times$ $10^6$ Mg of carbon storage. Under the scenario of natural development and urban development, the carbon storage of CCUA decreased by 6.21 $\times$ $10^6$ Mg and 5.41 $\times$ $10^6$ Mg, respectively, compared with 2020, mainly due to the reduction in grassland area and the increase in built-up land. Under the ecological protection scenario, land use change was relatively small, the downward trend of carbon storage was reversed, and a relatively stable situation was maintained.

## 4. Discussion

### 4.1. Response Relationship between Carbon Storage and Land Use Change

In CCUA, with the development of the social economy, and the population growth and land policy implementation of the government, the growth rate of built-up land was the highest from 2010 to 2020. In 2020, the permanent population of Chengdu and the central urban area of Chongqing was 31.27 million. Compared with 2010, the population increased by 9.76 million, with a population growth rate of 45.37%. The continuous inflow of the population has brought about social needs such as employment and housing. It has also promoted economic growth and accelerated urbanization [26]. At the same time, due to the increasing demand for built-up land, agricultural land and ecological land have inevitably been occupied, which has had a huge impact on the carbon storage of terrestrial ecosystems [40]. The research results also confirmed that the carbon storage of CCUA showed a downward trend from 2010 to 2020. The trend of land use change and the prediction of carbon storage in 2030 showed that ecological protection policies have strong impacts on the future carbon storage of CCUA. According to the calculation results of carbon storage over the years, farmland was the largest carbon pool among all land use types, followed by forest land. Therefore, the protection of farmland and forest land plays a decisive role in its future carbon storage.

It is noteworthy that the country's future development policy prioritizes ecological safety and takes ecological environment protection as the premise of economic development [42,43]. When simulating carbon storage under the three scenarios of CCUA in 2030, the scope of nature reserves was taken as the restricted area of land use conversion. Therefore, the research results show little difference in carbon storage under the three scenarios of natural development, ecological protection, and urban development. Even under the ecological protection scenario, the carbon storage increment was small, which does not mean that ecological protection is meaningless. In 2030, the total carbon storage of CCUA will show a downward trend under the natural and urban development scenarios. Therefore, the trend of carbon storage decline cannot be reversed if the focus is only on the protection of nature reserves, and if natural development is allowed in other regions without any intervention, which also reflects the importance of increasing ecological protection.

### 4.2. Advantages and Limitations of the Model

At present, research on land use and carbon storage generally adopts model simulation. The spatio-temporal change in carbon storage has been studied according to the mutual conversion between different land types [7,44]. In addition, integrated models have been used to consider the increase in regional carbon storage into the optimization process of land use [15]. This study estimated the carbon storage of CCUA by combining the FLUS and InVEST models. Nature reserves are also regarded as restricted development areas, which is more in line with the trend of land use change in the future. The FLUS model couples the top-down prediction model and the bottom-up CA model. It has effectively improved the simulation accuracy of land use patterns, and is widely used in simulations of land use patterns under different scenarios.

In the InVEST model, the carbon storage of ecosystems is calculated using the carbon module based on land use data and carbon pool [14]. It can calculate the carbon storage and distribution of each type of land use in a rapid and efficient manner. It is ideal in terms of analysis function, application cost, and evaluation accuracy, and has the advantages of simplicity and efficiency. The accuracy of the InVEST model has also been verified through many achievements in simulating the change in carbon storage in terrestrial ecosystems. However, the carbon density of each land use type is taken as a constant in the carbon pool table, although the carbon density of different vegetation cover and soil types may significantly vary. Consequently, the spatial heterogeneity of land use types is not taken into consideration. Moreover, the model does not take into account interannual changes in carbon density, which will increase the uncertainty of the results. According to previous research results, the annual change in regional carbon density data is small and has no significant impact on the estimation of the carbon storage of large-scale terrestrial ecosystems. In future research, the accuracy of the model in estimating carbon storage can be substantially improved by using carbon density data based on interannual measured data and distinguishing different vegetation and soil types in more detail. In this study, the mean value of the sample points inside and around the study area was taken as the carbon density data to minimize errors as much as possible. Although the model has some shortcomings, its results clearly reflect the spatio-temporal change in carbon storage in CCUA from 2010 to 2020 and in 2030 under multi-objective scenarios.

*4.3. Suggestions for Future Land Use Planning*

Government policies are mandatory for land use conversion, and the introduction of relevant government policies has played a guiding role in land use change [34]. More than 60% of the land in the study area was farmland. Urban expansion is the most important reason for the reduction in farmland in the Chengdu–Chongqing region, and its impact on farmland is irreversible. As a major grain-producing area in the western region, it is of great significance to safeguard food security. The government should ensure that the quantity and quality of farmland are not reduced in urban construction planning, and the agricultural industry is reasonably arranged, actively building a higher level of Tianfu granary. Farmland is the most important component of the carbon storage of CCUA. The protection of farmland plays an important role in enhancing the carbon sink capacity of the terrestrial ecosystem in the region. Therefore, in terms of land use planning, the approach of ensuring both quantity and quality of farmland should be followed. This also follows the current approach of three zones delineated by three lines for land use in China. The three zones are agricultural zones, ecological preservation zones, and urban development zones, while the three lines are the red line for protection of farmland and permanent basic cropland, the red line for ecological conservation, and the boundary line for urban development [45]. Delineating the red lines for the protection of farmland and permanent basic cropland, as well as the red lines of ecological protection, is especially beneficial for increasing carbon sequestration capacity. The main role of built-up land is as a carbon source, and its carbon fixation capacity is weak. Therefore, in the process of urban development, the total amount of built-up land should be effectively controlled.

**5. Conclusions**

This study combined the FLUS and InVEST models and simulated the land use pattern of CCUA in 2030 under multi-objective scenarios using land use data covering 2010–2020. The InVEST model was used to analyze the carbon storage and explain the potential impact of land use on the carbon storage of CCUA. The empirical conclusions are as follows: (I) The FLUS model has high accuracy in predicting the land use type of CCUA. The overall accuracy of the model was 91.18%, and Kappa coefficient was 83.71%, indicating a good performance for predicting the spatial pattern of future land use. (II) The main land use types of CCUA are farmland and forest land. From 2010 to 2020, the area of farmland decreased significantly (1422.83 km$^2$), and the area of built-up land increased

sharply (1401.70 km$^2$). The primary pathway of land use transfer was the conversion of farmland into built-up land. Under the ecological protection scenario, each land use type showed the smallest range of change, and the comprehensive land use dynamic rate was only 0.19%. Under the natural development scenario, the area of built-up land, wetland, and forest land will increase in 2030. Under the urban development scenario, the area of built-up land will increase by 751.24 km$^2$, accounting for an increase of more than 10.08%. In contrast, the areas of farmland, forest land, and grassland will decrease. (III) Land use has an important impact on carbon storage, and the changes in land use patterns affect the changes in carbon storage. The spatial pattern of carbon storage is "high in the east and west, low in the middle". In particular, farmland is the largest component of carbon storage in CCUA, accounting for over 60% of the total carbon storage. Carbon storage decreased by $29.45 \times 10^6$ Mg from 2010 to 2020. Grassland showed the most significant decrease in carbon storage, decreasing from 7.49% in 2010 to 6.09% in 2020. In 2030, under multi-objective scenarios, the total carbon storage will reach $1844.68 \times 10^6$ Mg under the ecological protection scenario, which is slightly higher than that in 2020. Under the natural development and urban development scenarios, carbon storage shows a downward trend.

**Author Contributions:** Z.S. and Z.L. were responsible for the design of research ideas. Y.L. and G.L. collected and analyzed the data, and J.C. and C.C. helped the format correction. Z.S. wrote the paper. All authors have read and agreed to the published version of the manuscript.

**Funding:** This work was supported by the scientific and technological research tasks of the Sichuan Academy of Agricultural Sciences (1+9KJGG009), the Sichuan Province Distinguished Youth Scholar Project (2020JDJQ0073), the Sichuan Province Key Research and Development Plan (2021YFYZ0028), the Sichuan Provincial Financial Independent Innovation Project (2022ZZCX036), and the Sichuan Province Science and Technology Planning Project (2022JDR0172).

**Data Availability Statement:** The raw data supporting the conclusions of this article will be made available by the authors, without undue reservation.

**Acknowledgments:** The authors acknowledge the data support from the Resource and Environment Science and Data Center, the Chinese Academy of Sciences (https://www.resdc.cn/, accessed on 10 December 2022), the National Meteorological Science Data Center, the National Science & Technology Infrastructure of China (http://data.cma.cn, accessed on 10 December 2022), the Open Street Map (https://www.openstreetmap.org/, accessed on 12 December 2022), and the Geospatial Data Cloud (http://www.gscloud.cn, accessed on 10 December 2022). We would like to thank the anonymous reviewers for their valuable comments and suggestions. We would like thank Yang Huang and Xuan Luo for providing language and graphic proofreading.

**Conflicts of Interest:** The authors declare no conflict of interest.

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
