# Peer review of "Impact of Land Use Change on Carbon Storage Based on FLUS-InVEST Model: A Case Study of Chengdu–Chongqing Urban Agglomeration, China"

_land, doi:10.3390/land12081531_

Round 1
Reviewer 1 Report
The authors studied the impact of land use change on carbon storage based on the FLUS-InVEST model: a case study of the Chengdu-Chongqing urban agglomeration, China. Since 2011, The Chengdu-Chongqing urban agglomeration has been a metropolitan area recently focused on by the Chinese government, with a large economic volume and rapid development. The study of its land use change and its carbon stock change is meaningful for the sustainable development of the region. However, the following issues still exist in the manuscript that needs to be focused on:
1. Firstly, in the abstract section, for example, the word 'Chengdu-Chongqing urban agglomeration' appears five times in 331 words. It is recommended to review and streamline it. Meanwhile, the same reason in the following text should be recognized and streamlining.
2. About data sources, you describe that Land use data is produced using Landsat TM/ETM remote sensing images as the main data source, but the Landsat TM sensor stopped working in November 2011, officially retired in January 2013 and the ETM sensor malfunctioned in 2003 (imagery with bad strips). However, your data sources cover 2015, 2018, and 2020, which seems to not match. Please check.
3. The font format in the figure is not consistent. For example, the font format in Figures 1 and 2.
4. The WEB of the dataset should be accompanied by the date of access. For example, Table 1 and numbers in 147-148.
5. Because the China's western region are significantly different from those in eastern China in terms of landform and climate. I think that carbon density data are subject to revision. In the subsections 2.3.4, author needs to explain this.
6. Section 3.1 of the article should belong to the section on research methods. I suggest that put it in section 2.3 of the article.
7. In Section 3.2, you mention that "In 2020, farmland accounted for 61.76% of the total land use in the region, while forest land accounted for 27.12%. Grassland and built-up land accounted for 5.24% and 1.75%, respectively. The area of water and other land use types is less than 2.00% of the land use area of the whole region." The figures total 97.78%. Is that right? Please check or explain it.
8. In Table 6, please indicate the years on the vertical and horizontal axes for easy identification.
9. In Section 3.3, using urban development in the text, but using town development in Table 7. The same condition occurs in the conclusion text.
10. The presentation of Table 8 in the article is incomplete.
11. In the subsections 4.2, because the model has a lot of research results in different areas, the advantages and disadvantages of the model are well known. This part of the article is more to elaborate what is already known. I think that the authors need to evaluate the applicability of the model in this region.
12. In section 4.3 of the paper, the author mainly expounds the impact of China's policies on the future development of land use planning. I think the author needs to discuss the future development of land use planning in this region under the regional policies of Sichuan Province and Chongqing Municipality.
13. The article uses three different descriptions of farmland, such as cultivated land, arable land, and farmland. Meanwhile, the forest land and built-up land have the same condition, such as woodland, and construction land. It is suggested that it should be noticed and described with one of these words.
14. More carefully, check the references in your article, such as reference 1, which is vol. 53. Meanwhile, I suggest citing more reference papers published in the journal Land (ISSN 2073-445X) in which you try to publish.

Overall, there are many repetitive terms in the writing of this article, such as Chengdu-Chongqing urban agglomeration. It is recommended to streamline it to make the article smoother. In addition, some keywords in the article may exhibit inconsistencies during translation, such as farmland, forest land, and built-up land. Please check.
Reviewer 2 Report
Dear Authors
Required edition:
1. On page 9 of Table 5. Reference values of land use carbon density in the study area, you need to add an additional column for total carbon storage.
2. Section 3.4. Spatial distribution characteristics and changes of carbon storage, after Figure 5, it is desirable to provide summary descriptive statistics in tabular form that clearly show the variability of carbon storage.
3. On page 14, lines 378 and 379. It is indicated that agricultural land accumulates more carbon. Due to what is this happening? Does the above-ground part of the plants remain in place? What types of breeding plants are grown? Maybe the NDVI index is taken in the maximum growing season of plants? As a rule, agricultural lands lose their carbon potential. If you do not apply mineral and organic fertilizers, or leave unclaimed parts of plants on the field after harvesting. Can you explain please.
4. It is necessary to edit lines 415-421, here one thought is given twice and almost identical sentences.
Accept after Minor Revisions: The paper can in principle be accepted after revision based on the reviewer’s comments.

Reviewer 3 Report
Line11,12to be rewritten to give correct meaning
Line20,21No mention of future scenarios
line 66-68 not clearly expressed
line 94-68 objectives to be brief and clear.
Fig1:Please check the river line in the two sub figures
Table1,use restricted in place of restrict
Climate environment may be replaced by climate topography
Social economy may be replaced by socio economical data
Table2: water may be replaced by wetland at land use 4 and accordingly in all places of its appearance
line 161 and explore .... and to explore
line167 model that simulates random process
para3.3 Model sensitivity to which parameters may be brought out
Uncertainty component in LULC prediction is missing
Fig4: Sub figures A, B, C may be described in figure title.
Fig5:Sub figures A, B, C, D may be described in figure title.
Fig6: Sub figures A, B, C may be described in figure title.
Social economy may be replaced by socio economical data
Line11,12to be rewritten to give correct meaning
Table1,use restricted in place of restrict
line 161 and explore .... and to explore
line167 model that simulates random process
Round 2
Reviewer 1 Report
The author made a good effort with most of the modifications based on the first review comments, but there is still only a small issue that should be further corrected. That is, the term of InVEST, I suggest that should be described like "Future land use simulation model (FLUS)", listing its full name is better. Meanwhile, list them as keywords that will be fit your subject. In addition, the term of CCUA in the keywords list should have its full name, such as Chengdu-Chongqing urban agglomeration (CCUA).

Author Response
We have made modifications. Provide full names for the FLUS model and InVEST model in the abstract. Add scenario and InVEST model as keywords. Provide full name for the Chengdu-Chongqing urban agglomeration in the keywords.